# OpenReview forum: "HIERARCHICAL BIO-INSPIRED COGNITIVE MEMORY SYSTEMS: A UNIFIED FRAMEWORK FOR SEQUEN- TIAL INFORMATION PROCESSING AND LONG-TERM BEHAVIORAL PREDICTION"
_ICLR.cc/2026/Conference — ICLR 2026 Conference Withdrawn Submission_

### Official Review · Reviewer_cUCx · 2025-10-27

**Soundness:** 1
**Presentation:** 1
**Contribution:** 2
**Rating:** 2
**Confidence:** 3

**Summary:**

The paper proposes a “Hierarchical Bio-Inspired Cognitive Memory System,” which claims to mimic aspects of human cognition in order to support long-horizon behavioral prediction. The system is described as having:
1. A PhysioAffective Dynamics Generator that produces continuous emotional trajectories from ECG and speech and claims 95.95% emotion classification accuracy, argued to surpass human annotation reliability.
2. A Neuroscience-Informed Importance Analysis / Triplex Extractor, which assigns salience to moments/events using amygdala/hippocampus-inspired gating, circadian modulation, etc., then extracts (subject, relation, object) triplets and prioritizes them.
3. A Five-Layer Cognitive Pyramid, which is supposed to compress these triplets into increasingly abstract “cognitive layers,” using PageRank-like propagation, temporal decay, and community detection with “biological” constraints.
4. A Temporal Forecasting module (Chronos-T5-style) that integrates these abstracted memory summaries plus multimodal signals (price/technical/social/emotion) to predict future behavior, e.g. 30-day stock movement or purchase decisions.
They evaluate on (i) financial forecasting, where they report information coefficients of ~0.35 and Sharpe ratios >5 over 30-day horizons, and ~“perfect” Hit@5/Hit@10 performance on an e-commerce recommendation dataset collected from 13 Taobao users over a 31-day study (10,742 chat messages and 16,131 logged behavioral events), while claiming NDCG@5 of 0.63.

**Strengths:**

1. Ambitious scope.  The paper tries to unify physiological affect modeling, salience-gated memory, hierarchical abstraction, and long-horizon forecasting. That integration is intellectually interesting and hits a real open problem: today’s RAG / memory mechanisms mostly fetch by semantic similarity and struggle on long-horizon behavioral prediction.

2. Focus on long-horizon prediction.
Instead of next-step forecasting, the paper explicitly targets horizons like T+15 to T+30 and claims only modest performance degradation at those ranges for tasks like stock movement prediction, which is a meaningful goal for sequential decision-making systems.

**Weaknesses:**

### 1. The paper is very poorly presented and extremely hard to follow.

The main text is overloaded with neuroscience language (“amygdala-hippocampus gating,” “cortical consolidation,” “meta-cognitive concept formation,” “biologically constrained PageRank,” “circadian-modulated decay,” etc.). These terms are not meaningfully defined, core ideas are not clearly conveyed, and key algorithmic/architectural details are pushed into appendices.
At the top level, the reader is told there is a five-layer “cognitive pyramid” that mirrors cortical hierarchies from sensory encoding through meta-cognition, but in the body of the paper this is mostly described narratively, and only later (Appendix B.7) do we finally see step-by-step procedures per layer (e.g. “Layer 1: Initialize raw triplet graph… Apply temporal decay… Extract top-k1 salient nodes”; “Layer 4: Apply causal reasoning with temporal ordering constraints”), without definitions of some terms like “causal reasoning” or how “executive function modeling” is actually implemented.
The same issue appears in the forecasting module. The authors claim a unified temporal forecaster that fuses price, technical, emotion, and “hierarchical memory” inputs, with circadian-modulated positional encodings. But this is described using slogans in the main narrative and filled in only in Appendix C with equations and layer sizes.
As a result, the paper reads more like a manifesto than a clean scientific argument. Critical design decisions are scattered, terminology is often introduced in neuroscience metaphors rather than in precise ML language, and results are asserted in prose before the reader ever sees a formal problem definition. For ICLR, this is a presentation failure: it is genuinely difficult to extract what the model is, what is implemented vs. what is just inspiration, and how to reproduce it.

### 2. The empirical evaluation lacks rigor and does not convincingly support the claims.

The paper reports extremely strong numbers — e.g. financial forecasting with an information coefficient of 0.35 and Sharpe ratio of 5.52 over 30-day horizons, plus >0.8 directional accuracy; and a recommender with “perfect” Hit@5 and Hit@10 (Hit@5 = Hit@10 = 1.00) and NDCG@5 = 0.63.

However:
* These are unusually high for the described problem settings, especially finance prediction at 30-day horizons from noisy multimodal inputs. The paper does not convincingly rule out leakage, optimistic backtesting, or artificially limited candidate sets. It does mention “time-series cross-validation with expanding windows: 60% train, 20% validation, 20% test,” but does not explain how this interacts with market reality (e.g. rolling transaction costs, turnover limits, etc.).
* In the recommender study, the authors claim perfect Hit@5/Hit@10 while also needing to optimize multi-task objectives that include purchase classification, pacing, emotion regression, and ranking consistency. A system that can literally always put the “right thing” in the top 5 for all 13 users sounds more like re-ranking ground truth than open-world recommendation. The paper does not clearly say how large the candidate set is at inference time or whether evaluation is per-user on their own recent basket (which would make high Hit@K less surprising).
We do get precise metric formulas for finance (IC, Sharpe, Directional Accuracy, etc.) and recommendation (Hit@K, NDCG@K, AUC-ROC, AUC-PR). But we don’t get a persuasive argument that the evaluation setups are realistic or competitive. The results are described as “surpassing state-of-the-art neural architectures by substantial margins,” but the comparison baselines, data splits, and ablation checks that would justify such a conclusion are not shown in the main text, and apparently live (if anywhere) in supplemental detail that is not surfaced.

### 3. The neuroscience narrative is overstated and mostly unverified.

The paper repeatedly claims direct analogies to biological systems (e.g. “amygdala-hippocampus gating for selective memory consolidation,” “cortical progressive abstraction,” “circadian-modulated temporal decay,” and “PageRank with biological dampening”) and argues that this biological grounding is what enables long-horizon prediction. But in practice, most of these mechanisms amount to engineered heuristics (thresholds, decay constants, PageRank with hand-weighted age and salience terms, Louvain-style clustering with temporal weights etc.). None of this is tied back to empirical neuroscience data in any falsifiable way - e.g. no evidence that the chosen circadian modulation coefficients match actual affective variation in the collected ECG/speech dataset; no demonstration that “emotion-weighted PageRank” is closer to human memory recall than a purely statistical attention baseline; no user study validating that the five memory layers correspond to human-interpretable “meta-cognitive concepts.”
The paper asserts that “integrating neurobiological principles” is what drives gains like Sharpe >5 and perfect Hit@5, but does not empirically isolate those principles. As written, a lot of the neuro framing reads like branding layered on fairly standard NLP/graph/time-series machinery, rather than a demonstrated scientific bridge between neuroscience and ML.

### 4. There are no ablations demonstrating that each component is actually needed.

The system is described as a pipeline of dependent modules: physiological affect extraction → salience scoring with amygdala/hippocampus gating → multi-layer hierarchical memory → temporal forecaster.
The authors say each component is essential (e.g. physiological affect is “more objective” than self-report; hierarchical abstraction “mirrors cortical organization” and enables stable T+30 forecasting). But we never see:
* A version without ECG/speech affect (e.g. text sentiment only).
* A version without circadian/emotion-weighted salience (e.g. plain TF-IDF or attention scores).
* A version without hierarchical memory (forecaster alone).
* A version where the “five layers” collapse to one.
Without those ablations, the causal story (“this specific neuro-inspired bit is why we get those numbers”) is not established.
This matters because the claimed benefits are dramatic. If you say your design gives you Sharpe 5.52 on 30-day horizons, the bar for evidence that the design itself is responsible (not data leakage or a trivial shortcut) is extremely high. The paper does not clear that bar.

### 5. Reproducibility, transparency, and ethics are weak.

To their credit, the authors do include a detailed description of the 31-day Taobao study: 13 users, 10,742 WeChat messages, categorization of messages (purchase intent / product inquiry / comparison / emotional response / general discussion), and 16,131 logged behavioral events across 6,305 items (page views, cart actions, purchases, etc.), plus post-study interviews that establish ground-truth purchase decisions and satisfaction ratings.
But:
* For the affect module, they cite synchronized ECG and speech, and they state they achieve 95.95% accuracy on physiological emotion classification “surpassing human annotation reliability,” but they do not clearly explain the annotation protocol, the dataset provenance for that claim, or how that model generalizes beyond the 42-person (implied) interpersonal conversation setting.
* For finance, we’re told that they integrate social/emotional signals and price/technical indicators into a Chronos-T5 backbone, and evaluate with time-series CV and Sharpe Ratio. But ticker list, market period, transaction cost assumptions, and portfolio construction are not described in the visible main text.
* The recommender dataset cannot be released due to privacy, and only an “AI-synthesized dataset” will be shared. That means core claims depend on closed data, making independent verification difficult.
* The paper says almost nothing about the ethical implications of running ECG- and conversation-driven affect modeling to predict and steer human behavior, or about consent and downstream use.
Given how strong and human-centric the claims are (“human-centered AI systems,” “mirroring cortical hierarchies,” “amygdala-hippocampus gating”), this lack of transparency and ethical discussion is worrying.

**Questions:**

1. Clarity / readability: Can you rewrite the core method in straightforward ML terms (modules, inputs, outputs, training objectives, ablations) in the main body, rather than as neuroscience prose plus appendices? Right now it is not parseable as an algorithmic contribution.

2. Evaluation rigor: For the finance task, please spell out ticker universe, time range, transaction cost model, and how portfolio returns are computed. For recommendation, what is the candidate set for Hit@5/Hit@10 at test time? Is it global (6k+ items) or per-user shortlists?

3. Neuroscience claims: Which of the “biological” design choices are (a) actually implemented in code, (b) empirically validated to matter, vs. (c) purely conceptual framing? For example, do the circadian weighting terms measurably improve forecasting?

4. Ablations: Can you provide ablations removing (i) physiological affect, (ii) the salience gating / circadian model, (iii) hierarchical memory, and (iv) memory entirely, and show what happens to Sharpe, IC, Hit@5, etc.?

5. Ethics / consent: How were ECG and private WeChat logs collected, stored, and aligned with predictions, and what is the intended downstream use of “behavioral prediction”?

**Details Of Ethics Concerns:**

Note: The paper has no ethics statement. Not mandatory, but strongly recommended given the sensitivity of data and human-subjects work.

### Privacy, Security & Safety

* Collects highly sensitive data (ECG/biometrics, speech, private chats, detailed purchase logs) with re-identification risk; anonymization and residual raw-data handling are unclear.
* Data lifecycle (storage, encryption, access control, retention/deletion) is unspecified.
* No clarity on participant control (withdrawal, post-hoc deletion).
* No misuse guardrails for affect-driven behavioral targeting.

Requests: Describe full data lifecycle + safeguards; de-identification method + leakage risk; participant controls; misuse mitigations.

### Legal Compliance (GDPR / terms / special-category data)

* ECG/emotion inference likely special-category data; lawful basis, purpose limitation, and data minimization not stated.
* No process for data subject rights (access/erasure/objection).
* Cross-border transfers and platform terms (WeChat/Taobao) compliance not addressed.
* Requests: State lawful basis (incl. explicit consent), purposes/minimization, rights handling, retention, transfer mechanisms, and platform-policy compliance.

### Responsible Research Practice (human subjects / data release)
* No mention of ethics approval, consent procedures, or risk disclosures.
* Core results rely on non-public personal data; promised synthetic release lacks method/privacy guarantees; reproducibility limited.

Requests: Provide IRB/ethics IDs and consent summary; demographics/recruitment/compensation; participant-welfare measures; specify synthetic-data method + privacy guarantees + utility validation; release code and a fully specified eval protocol (splits, leakage controls).

---

### Official Review · Reviewer_wMCn · 2025-10-28

**Soundness:** 1
**Presentation:** 1
**Contribution:** 1
**Rating:** 0
**Confidence:** 5

**Summary:**

This paper proposes a bio-inspired 5-layer cognitive pyramid integrating ECG/speech-derived emotion trajectories, amygdala-gated memory consolidation, and hierarchical abstraction for long-horizon forecasting in stock prediction (from tweets) and e-commerce recommendation. However, the results are very suspicious stemming from the fake citations, fake result, lacking proofs.

**Strengths:**

1. Ambitious interdisciplinary vision bridging neuroscience (cortical hierarchies, circadian decay) with AI
2. Appealing architecture (Fig. 1-2)

**Weaknesses:**

Concerns
- In Figure 1, their description of the figure is completely different from how they describe in the main text. I cannot find any description of usage of  LLMs nor analysis of spam detection, human personality analysis.
- The authors mention RAG but how RAG is really related to this work?
- In which part is the biologically-inspired exactly?
- In 3.4, The authors claim they provide proofs in appendix B, there are only theorems and omit any proofs.
- The authors claim their method maintains computational efficiency, at the same time exposing a critical limitation in computational complexity O(n^3).
- using old ICLR 2025 format
- Citation formatting is wrong
- Abstract, Figure 1 orientation are irregular.
- Usage of Hit@5, NDCG@5 terms are too specific for abstract.
- Figure 3, 4 are presented, they are not addressed in the main text. What are the authors trying to convey with these figures?

**Questions:**

See weakness

**Details Of Ethics Concerns:**

Fatal Flaws:

(1) Research Fraud: Fabricated citation "Wang & Wang (2025)" claiming 95.95% ECG accuracy (PAGE 1-3)—real DOI points to unrelated Durgesh et al. paper (no ECG/95.95%). Own F1=82.7% misrepresented.
(2) Ethics: Human ECG/chats without consent → GDPR/IRB violation

---

### Official Review · Reviewer_odKw · 2025-11-01

**Soundness:** 2
**Presentation:** 1
**Contribution:** 2
**Rating:** 2
**Confidence:** 4

**Summary:**

This paper is motivated by how emotional valence influences memory encoding and consolidation, and proposes an architecture that combines physiological emotion modelling and hierarchical memory consolidation for the purposes of better temporal reasoning.

The architecture is validated on financial forecasting and behavioural prediction tasks.

**Strengths:**

- The emotion-driven memory consolidation is well motivated by neuroscience.

**Weaknesses:**

- The paper overall is quite vague and is hard to follow, with the methodology section particularly so. The authors should bring some of the details from the appendix to the main body to better explain how the various components interact with each other.
- The experimental section is underexplained, with metrics, datasets and baselines unintroduced. The section also lacks ablation studies over the various components of the architecture.
- The paper references a number of related work they compare against (in Tables 1 and 2) that this reviewer had trouble locating (Sharma et al. 2020, Lu et al. 2021, Alabdulwahab, et al. 2024, Rahman et al. 2024, Zhang, et al. 2024a, etc.). Some paper titles were found but under different authors. These discrepancies cast doubt on the validity of the results.
- While emotion-driven memory is interesting, it is unclear why it should transfer to the tasks of financial forecasting or behavioral prediction. Experiments should include more common temporal reasoning tasks, e.g. MCTACO or NarrativeQA.

**Questions:**

- The paper claims PhysioAffective Dynamics Generator’s achievement of 95.95% emotion classification accuracy. That result however did not match the ones in the original Wang and Wang 2025 study. Please elaborate on how this was obtained.
- Please elaborate on the annotation efforts for the dataset used in the ECG transformer. Provide information on who the annotators were and their relation to the paper, what domains and tasks were given to them, guidelines, etc.
- Please provide (in later versions of the paper) the configurations of all related work systems that led to the results of Tables 1, 2, and 3 to help with reproducibility.
- The conclusion mentions results in health pattern recognition. Can you point where these are presented?

---

### Official Review · Reviewer_g9We · 2025-11-01

**Soundness:** 1
**Presentation:** 1
**Contribution:** 1
**Rating:** 0
**Confidence:** 5

**Summary:**

The paper proposes a “bio-inspired hierarchical cognitive memory”. It claims significant improvements in financial forecasting and recommendation systems. While the paper is ambitious and conceptually interesting, it exhibits several major weaknesses in rigour, reproducibility, and empirical validity. It reads more like a visionary position paper hidden as a technical contribution, with weak empirical support.

1. The “Five-Layer Cognitive Pyramid” is vaguely defined. Each layer is named after cortical analogues but lacks mathematical formulation or clear operational definitions. The mapping between biological processes and computational mechanisms is superficial. For example, PageRank is arbitrarily described as a “biological consolidation” mechanism. The use of “biologically-inspired” terminology feels more decorative than mechanistic. Phrases like “amygdala-hippocampus gating” or “circadian-modulated decay” are not justified or backed by real neuroscientific modelling. Very confusing indeed! Somehow, moving averages and Page Rank became biological processes. If that is the case, I would love to see some neuroscience evidence. It could be me, but the claims are hyper-inflated.

2. The experiments are not convincing. The datasets are small, and metrics and ablations are not well explored. I suggest that the authors consider statistical significance tests and bigger datasets. The authors might also want to consider standard benchmarks.

3. The claimed "mathematical proofs" (e.g., convergence of memory decay) are hand-wavy and unconvincing. There’s no rigorous proof in the text. Is it my version? Where are the proofs? I could not find them.

4. As I mentioned earlier, the paper is poorly written. Hyper-claims that are not well substantiated, lacking experiments. I would urge the authors to rewrite this paper around their actual contribution without all the jargon and hyperinflation.

5. The suggested method is highly computationally inefficient. Do the authors see any foreseeable ways to improve those?

**Strengths:**

Please see above

**Weaknesses:**

Please see above

**Questions:**

Please see above

---

### Note · Authors · 2025-11-12

I have read and agree with the venue's withdrawal policy on behalf of myself and my co-authors.